# Bioelectrical impedance analysis for early screening of upper limb subclinical lymphedema: A case–control study

**Linli Zhuang**[☯], **Huaying Chen**[☯], **Xuemei Zheng**[☯], **Shaoyong Wu**[ID]*, **Youhui Yu, Lu Lan, Liang Xu, Jumei Xu, Hongying Fan**

Department of Head and Neck Oncology, Cancer Center, West China Hospital, Sichuan University, Chengdu, China

☯ These authors contributed equally to this work.
* wusy6@163.com

**Data Availability Statement:** All relevant data are within the manuscript and its Supporting Information files.

## Abstract

Breast cancer-related lymphedema is a treatment-related chronic disease that causes great distress and medical burden. Early screening and precautionary measures for lymphedema could improve well-being and decrease medical costs. Herein, we used bioelectrical impedance analysis for early screening of lymphedema. We set up a verifiable standardized subclinical standard to screen subclinical lymphedema in postoperative breast cancer patients using bioelectrical impedance. The first part determined the criteria of subclinical lymphedema. Among the 424 female participants, 127 were healthy women, whereas 297 were postoperative breast cancer survivors. Subclinical standard boundaries were determined by the 95% confidence interval of the healthy women. The screening rate of patients with subclinical lymphedema was inferred by comparing the subclinical standard boundaries and the postoperative patient values. A total of 14.81–20.87% of postoperative breast cancer survivors were identified as patients with subclinical lymphedema. The second part provided the results of the verification test of this subclinical standard. The data of the verification test from 30 healthy women and 30 screened patients met the subclinical standard, and 30 breast cancer survivors with lymphedema verified the utility and feasibility of the subclinical standard. Therefore, this standard could provide a screening tool for early the identification of subclinical breast cancer survivors. Early detection helps implement personal and precise medical precautions for patients with subclinical lymphedema.

## Introduction

Breast cancer-related lymphedema (BCRL) is a treatment-related chronic disease without a radical cure [1, 2]. This disease has an insidious onset and progressive chronic course and imposes a heavy disease burden on breast cancer survivors [3, 4]. A previous study of 2171 American breast cancer survivors with a 5-year follow-up revealed that 13.7% of the survivors had upper limb lymphedema following surgical treatment [5]. In China, up to 49% of patients

**Funding:** The author(s) received no specific funding for this work.

**Competing interests:** The authors have declared that no competing interests exist.

who undergo mastectomy develop lymphedema [6]. The discrepancy is probably traceable to the extent of lymph node removal, study design, different populations, self-management ability, lifestyles, etc. [7, 8]. Furthermore, breast cancer survivors with lymphedema not only experience skin changes, progressive swelling, and pain in the arms but also have restricted arm movement. BCRL often coexists with psychological manifestations such as self-image disturbances, fear, depression, and lower life satisfaction [9]. Therefore, early identification can promote early protective intervention, preventing lymphedema from progressing and improving the quality of life and well-being.

Reliable and valid standards are the core of early detection. The utility and validity of the subclinical standard relies on the ability to distinguish patients with subclinical lymphedema from postoperative patients. In fact, in the early stage of lymphedema, the absence of swelling and volume measurement makes it challenging to assess subclinical lymphedema [10]. Self-assessment acts as a supplemental screening tool, contributing to the diagnosis of secondary lymphedema but not subclinical lymphedema [2, 11]. However, continuous monitoring using bioimpedance spectroscopy can help in the early detection of lymphedema, and early intervention for high-risk patients can further reverse or delay the course of lymphedema [10]. Thus, bioelectrical impedance (BI) measurement is well-adapted for the early screening of subclinical lymphedema.

BI analysis (BIA) using biophysical model-based algorithms can directly evaluate the impedance and resistance of the body. It also helps assess body composition indirectly by transforming the value of BI into the value of body composition [12]. However, there is no standardization of assessment protocols for subclinical lymphedema. The BI value varies between different device models and versions [13, 14]. It is therefore, essential to address this issue. Our study aimed to establish a subclinical standard that can screen high-risk patients for upper limb lymphedema early using BIA. This subclinical standard should possess the characteristics of clinical utility and standardization. It would be best to establish a uniform subclinical standard among different bioelectric device models and versions. This standard can be validated by using real-world clinical data. Moreover, a previous study has implicated trauma to the lymph nodes as a fundamental reason for BCRL [15]. However, there is a paucity of evidence regarding the correlation between subclinical lymphedema and the extent of nodal surgery. Our study attempted to directly assess the possibility that trauma to the lymph nodes mediates the risk of secondary lymphedema using a BI device. This study aimed to establish and verify a subclinical standard of BI and to evaluate whether breast cancer survivors with trauma to the lymph nodes had a higher risk of lymphedema. This study aimed to evaluate whether subclinical lymphedema screening assessment in breast cancer survivors in the early postoperative period using BI resulted in a more accurate identification of subclinical lymphedema caused by nodal surgery.

## Materials and methods

### Participants

This study was a case–control trial conducted at West China Hospital (a large district general hospital) between January 2021 and December 2021. The participants' data were collected during the past year. A total of 127 healthy female adults were enrolled in the study. A total of 297 patients who underwent unilateral breast and lymph node removal were included in the screening group. Six patients were excluded as they had bilateral breast cancer. Each trial participant provided written informed consent. In this study, BIA was performed on both healthy controls and patients.

Healthy females aged 18–55 years were included in the study. Healthy women were included in the control group. After surgery, 297 eligible patients with breast cancer were assigned to the screening group. In China, doctors recommend the initiation of chemotherapy ≤30 days after surgery. Postoperative patients underwent the BI measurements on the day before the first chemotherapy session. Breast cancer survivors who received intravenous chemotherapy or radiotherapy had significantly changed BIA results. The exclusion criteria were as follows: (1) participants with a history of radiotherapy and chemotherapy; (2) participants diagnosed with kidney disease, cardiovascular events, lymphedema-related diseases, and immune disorders; (3) participants who were menstruating or pregnant during the analysis.

## Demographic characteristics and BI measurement

Implementation of effective and timely guideline-recommended screening in breast cancer survivors is needed for early identification of subclinical changes in lymphedema. Ordinarily, lymphedema-related changes in BI devices are detected earlier. Early detection of secondary lymphedema implies early treatment and management. Given the ability for early detection of lymphedema in patients with breast cancer, BIA was used in American centers as a screening tool for breast cancer survivors after axillary surgery [16]. BI measurement is a quick test to screen breast cancer patients at risk of lymphedema. Before each measurement, demographic data on age, sex, marital status, date of birth, ethnicity, and residential region were collected through brief conversations. For the patient group, treatment-related data (basic medical history, surgical history, and date of breast cancer surgery) were also collected. A skilled therapist in our study assessed several aspects before measurement, such as whether the inclusion and exclusion criteria were met, whether skin changes occurred, whether limb pain or discomfort existed, whether the patient experienced upper extremity swelling or nonpitting edema, and whether the range of motion was restricted. BIA data were collected by a trained therapist. The therapist wiped the electrodes using alcoholic tissue before every measurement and steered the participants who used light clothes through this measurement to increase the accuracy of the measurement. The time spent on this process was less than 10 min. BI measurements were performed in the morning after overnight fasting and bladder emptying. All participants were instructed to stand on the platform of an InBody 770 multifrequency BI device (InBody 770, Cerritos, CA, USA), with whole soles touching the voltage-sensing electrodes. The participants then stayed motionless as the device measured their body weight. In the subsequent step, the participants held the handles with the thumb in contact with the hand electrodes and kept the two arms in the right position. All the participants completed the BI measurements. All authors could access information that could identify individual participants during data collection and statistical analysis.

## Evaluation criteria

There is no international standard for screening subclinical patients. "Screened patients" were defined as those who fulfilled the criteria for subclinical lymphedema in our study. In oncology, arm volume increased more than 3% corrected for body weight is widely used as a definition for early onset lymphedema [17]. However, these standards apply only to clinical lymphedema cases and are less able to detect subclinic BCRL [18]. In our study, all data from the healthy participants were used to establish a 95% reference range. According to this range, the feasibility of each indicator was assessed using the screening rate (the number of screened patients divided by the total number of patients with breast cancer). The verification method was then used to select the sensitive indicators. Finally, subclinical screening standards were constructed using sensitive indicators.

### Verification data sets of subclinical standards

A systematic review and meta-analysis of 72 studies showed that the incidence of lymphedema in patients who underwent unilateral breast and lymph node removal varied between 13% and 21% according to multiple criteria (clinical assessment, arm circumference, and self-reporting) [8]. Within the scope of the current incidence, the five indexes presented in this paper are considered subclinical standards. To further verify this subclinical standard, we added a confirmatory assay including 30 healthy control participants, 30 screened patients meeting subclinical standards, and 30 lymphedema breast cancer survivors. The reference range for our standard should be validated in patients with lymphedema. Ideally, a few patients with lymphedema were missed by this standard, and the reverse applied to the healthy population. Fig 1 shows participant flow.

### Statistical analysis

All statistical analyses were performed using SPSS version 25.0, and statistical significance was set at $P \leq 0.05$. Student's t-test and analysis of variance were used to assess the differences in continuous variables between the control and patient groups. Given the sensitivity and specificity of the diagnostic thresholds, the threshold was set at two standard deviations above the mean of the healthy women to establish subclinical screening criteria [13]. Therefore, the reference range of each indicator was defined as the mean value plus or minus two times the standard deviation. High-risk lymphedema groups (screened patients) were defined as participants who had special measurement values beyond the threshold of the reference range.

Previous studies have noted the difference between dominant and nondominant limbs in terms of the impedance ratio. In fact, the difference between both arms not only exists in the dominant and nondominant arms but also in the proximal and distal arms (according to the distance between the breast cancer location and arm). These differences between the dominant and nondominant limbs (proximal and distal arms) could be addressed by evaluating the ratio difference in both upper arms. The ratio difference is given as follows:

Ratio difference
= absolute value of difference between both upper limbs/the minimum value of both upper limbs
× 100%.

## Results

### Demographic characteristics

A total of 424 participants were enrolled in our study from January 2021 to December 2021. All participants were females. The mean age of the control group was 39.2 years (range, 24–55 years), and the mean age of the patient group was 47.2 years (range, 27–75 years). Of the 297 patients with breast cancer in our study, 29 (9.8%) were older than the upper reference range of healthy women (65 years). In total, 424 participants had a height range of 137–172 cm in our study. Table 1 shows that the participants in the patient group had a significantly higher weight and body mass index (BMI) than those in the control group. Table 1 shows demographic and descriptive information.

### Fat-related characteristics of the control and patient groups

The normal reference range was defined as that of the control group (healthy women). The critical screening values of various parameters were determined using the normal reference

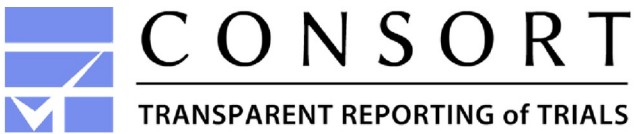

## CONSORT 2010 Flow Diagram

**Enrollment**

Assessed for eligibility (n= 430 )

Excluded  (n= 6 )
♦ Not meeting inclusion criteria (n=6)
6 patients was *excluded* due to the reason that patients suffer *bilateral breast cance*

**Allocation**

Allocation (n= 424)

Allocated to control group (n=127)
127 participants were healthy women

Allocated to screen group(n= 297 )
297 patients who removed unilateral breast and lymph nodes were included as screen group

**outcome**

1. established the subclinic lymphedema standard for screening the subclinic lymphedema patients after surgery

2. identified subclinic lymphedema patients

**verification**

confirmatory assay (n= 90)
♦Excluded from confirmatory assay (give reasons) (n= 0 )
30 healthy controls participants,
30 screened patients meet sub clinic standard,
30 lymph-edema breast cancer survivors

**Fig 1. CONSORT 2010 flow diagram.**

**Table 1. Demographic characteristics of the study population.**

| Variables | Control group (n = 127) | Patients group (n = 297) | t score | P score |
|---|---|---|---|---|
| Age | 39.15±13.28 | 47.15±12.27 | 9.665 | <0.001 |
| Gender(Female) | 127 | 297 | - | - |
| Height | 156.11±20.29 | 156.38±11.88 | 0.23 | 0.818 |
| Weight | 55.63±8.11 | 57.09±8.01 | 2.33 | 0.020 |
| BMI | 22.14±3.16 | 23.14±3.12 | 4.31 | <0.001 |

range and clinical judgment. In our study, more than eight indicators were related to fat. Participants in the patient group tended to have a higher proportion of fat than healthy women. In the current study, the mean weight of the control group was 55.63 kg (range, 47.52–63.74 kg). Of the patients included, eight (2.5%) weighed greater than 71.53 kg (the upper reference limit). Eighteen (6.06%) postoperative patients had a body fat mass higher than 27.91 kg/m$^2$. Twelve of the 297 (4.04%) patients had BMI values higher than the limit. Table 2 provides the fat characteristics of healthy individuals, including the percent body fat, obesity degree, fat-free mass (FFM) of the right arm, FFM% of the right arm, FFM of the left arm, and FFM% of the left arm. The critical values for percent body fat and obesity degree were 41.70 (screened patients = 21, screening rate = 7.07%) and 13.73 (screened patients = 9, screening rate = 3.03%), respectively. The critical values for FFM of the right arm and FFM of the left arm were 2.40 (screened patients = 10, screening rate = 3.36%) and 2.41 (screened patients = 7, screening rate = 2.35%), respectively. The screening rates of FFM% and FFM in the upper limbs were similar.

## Cellular water characteristics of the control and patient groups

Table 3 describes the total body water (TBW), intracellular water (ICW), and extracellular water (ECW) profiles of both groups. The upper reference ranges of the ECW/TBW of the right arm and ECW/TBW of the left arm were 0.3856 and 0.3867, respectively. The ECW/TBW of the unilateral arm tended to have a higher screening rate (screening rate of ECW/TBW of the right arm = 10.77%, screening rate of ECW/TBW of the left arm = 12.12%) among

**Table 2. Fat characteristics and screening rate.**

| Variables | Control group (N = 127) | The critical screening values (M±2SD) | The number of screened patients | screening rate (%) |
|---|---|---|---|---|
| Weight | 55.63±8.11 | 71.53 | 8 | 2.69 |
| Body Fat Mass | 17.19±5.47 | 27.91 | 18 | 6.06 |
| Soft Lean Mass | 36.16±3.79 | 43.59 | 5 | 1.68 |
| Body Fat Mass | 17.19±5.47 | 27.91 | 18 | 6.06 |
| Soft Lean Mass | 36.16±3.79 | 43.59 | 5 | 1.68 |
| Fat Free Mass(FFM) | 38.43±4.00 | 46.27 | 5 | 1.68 |
| Percent Body Fat | 30.37±5.78 | 41.70 | 21 | 7.07 |
| FFM of Right Arm | 1.79±0.31 | 2.40 | 10 | 3.36 |
| FFM% of Right Arm | 92.94±10.37 | 113.27 | 12 | 4.04 |
| FFM of Left Arm | 1.76±0.33 | 2.41 | 7 | 2.35 |
| FFM% of Left Arm | 90.85±10.89 | 112.19 | 9 | 3.03 |

*"Screened patients" were defined as those who fulfilled the criteria for subclinical lymphedema.

*"Screening rate" were defined as the number of screened patients /total number of patients*100%.

*"The critical screening values" The critical screening values of various parameters were determined using the normal reference range and clinical judgment.

**Table 3. Cellular water characteristics and screening rate.**

| Variables | Control group (N = 127) | The critical screening values (M±2SD) | The number of screened patients | screening rate (%) |
|---|---|---|---|---|
| Total Body Water(TBW) | 28.2±2.95 | 33.98 | 5 | 1.68 |
| Intracellular Water(ICW) | 17.36±1.84 | 20.97 | 4 | 1.34 |
| Extracellular Water(ECW) | 10.84±1.13 | 13.05 | 7 | 2.35 |
| TBW of Right Arm | 1.40±0.24 | 1.87 | 10 | 3.36 |
| TBW of Left Arm | 1.37±0.25 | 1.86 | 9 | 3.03 |
| ICW of Right Arm | 0.87±0.15 | 1.16 | 9 | 3.03 |
| ICW of Left Arm | 0.85±0.16 | 1.16 | 8 | 2.69 |
| ECW of Right Arm | 0.53±0.09 | 0.71 | 9 | 3.03 |
| ECW of Left Arm | 0.52±0.10 | 0.72 | 9 | 3.03 |
| ECW/TBW | 0.3842±0.0073 | 0.3985 | 10 | 3.36 |
| ECW/TBW of Right Arm | 0.3776±0.0041 | 0.3856 | 32 | 10.77 |
| ECW/TBW of Left Arm | 0.3789±0.0040 | 0.3867 | 36 | 12.12 |

*"Screened patients" were defined as those who fulfilled the criteria for subclinical lymphedema.

*"Screening rate" were defined as the number of screened patients /total number of patients*100%.

*"The critical screening values" of various parameters were determined using the normal reference range and clinical judgment.

all parameters of cellular water. The mean TBW, ICW, and ECW values were 28.20, 17.36, and 10.84, respectively. Only 4–7 patients with breast cancer (screening rate range, 1.34–2.35%) were screened by these indicators (TBW, ICW, and ECW). The number of screened patients and the screening rate were similar among the other variables of cellular water (screened patients = 8–10, screening rate = 2.69–3.36%).

## Bioelectrical analysis characteristics and screening rate

BI, reactance, and body angle analyses are presented in Table 4. The 50 kHz reactance and 50 kHz phase angle of the upper limbs had a higher screening rate. Similarly, the 5 kHz impedance and 50 kHz reactance had higher screening rates in the frequency range of 5 kHz to 1000 kHz. The critical value of 5 kHz impedance was 325.18 (5 kHz impedance in the right arm: screened patients = 11, screening rate = 3.70%) and 328.01 (5 kHz impedance in the left arm: screened patients = 13, screening rate = 4.38%), respectively. In total, 26 patients (8.75%) were selected using 5 kHz impedance. Furthermore, the critical values of the 50 kHz reactance were 21.92 (50 kHz reactance in the right arm: screened patients = 31, screening rate = 10.44%) and 21.18 (50 kHz reactance in the left arm: screened patients = 25, screening rate = 8.42%), respectively. In total, 56 (18.86%) patients were screened. Finally, the mean values of the 50 kHz phase angle were 4.5 (50 kHz phase angle in the right arm) and 4.26 (50 kHz phase angle in the left arm), respectively. The screening rate increased to 19.99%.

## Differential interlimb analysis

According to the current standards, clinical lymphedema is defined as the volume difference between the arms. Based on this principle, we established an indicator set for the interlimb difference. Data collected from the BI device included five parts: the left arm, right arm, trunk, left leg, and right leg. All the data are numerical-type variables. Therefore, an interlimb difference can be calculated between the values of the left and right arms using the same index. Previous studies have reported a difference between the dominant and nondominant arms. In fact, the difference between both arms exists not only in the dominant and nondominant arms

**Table 4. Bioelectrical impedance characteristics and screening rate.**

| Variables | Control group (N = 127) | The critical screening values (M±2SD) | The number of screened patients | screening rate (%) |
|---|---|---|---|---|
| 1 kHz-RA Impedance | 435.06±52.84 | 323.18 | 7 | 2.36 |
| 1 kHz-LA Impedance | 442.49±55.11 | 334.48 | 8 | 2.69 |
| 5 kHz-RA Impedance | 427.16±52.03 | 325.18 | 11 | 3.70 |
| 5 kHz-LA Impedance | 434.53±54.35 | 328.01 | 13 | 4.38 |
| 50 kHz-RA Impedance | 388.61±47.9 | 294.73 | 8 | 2.69 |
| 50 kHz-LA Impedance | 397.74±50.8 | 298.17 | 10 | 3.37 |
| 250 kHz-RA Impedance | 354.4±44.56 | 267.07 | 7 | 2.36 |
| 250 kHz-LA Impedance | 364.01±47.53 | 270.85 | 8 | 2.69 |
| 500 kHz-RA Impedance | 343.06±43.4 | 258.01 | 7 | 2.36 |
| 500 kHz-LA Impedance | 352.45±46.35 | 261.61 | 7 | 2.36 |
| 1000 kHz-RA Impedance | 336.72±42.7 | 253.03 | 7 | 2.36 |
| 1000 kHz-LA Impedance | 345.37±45.53 | 256.13 | 7 | 2.36 |
| 5 kHz-RA Reactance | 14.94±2.66 | 9.73 | 26 | 8.75 |
| 5 kHz-LA Reactance | 14.38±2.52 | 9.44 | 15 | 5.05 |
| 50 kHz-RA Reactance | 30.37±4.31 | 21.92 | 31 | 10.44 |
| 50 kHz-LA Reactance | 29.42±4.2 | 21.18 | 25 | 8.42 |
| 250 kHz-RA Reactance | 24.17±2.95 | 18.38 | 27 | 9.09 |
| 250 kHz-LA Reactance | 24.4±2.93 | 18.66 | 19 | 6.40 |
| 50 kHz-RA Phase Angle | 4.5±0.49 | 3.54 | 50 | 12.59 |
| 50 kHz-LA Phase Angle | 4.26±0.50 | 3.28 | 22 | 7.40 |
| 50 kHz-Whole Body Phase Angle | 4.86±0.51 | 3.86 | 20 | 6.73 |

*"Screened patients" were defined as those who fulfilled the criteria for subclinical lymphedema.

*"Screening rate" were defined as the number of screened patients /total number of patients*100%.

The critical screening values of various parameters were determined using the normal reference range and clinical judgment.

* LA: left arm RA: right arm

but also in the proximal and distal arms (according to the distance between the breast cancer location and arm). Our study showed that differences between arms exist in breast cancer survivors, which has not been suggested by previous research. As an exploratory study, the difference between both arms in the healthy group will be used to provide a benchmark that uses two standard deviation cut-offs for screening breast cancer survivors [17]. For example, the ratio difference of ECW was calculated using the absolute value of the ECW difference in both upper limbs divided by the minimum value of ECW of both upper limbs × 100%. All the inter-limb indicators adopted a similar formula for the calculation. The reference range of the ECW ratio difference, the FFM ratio difference, the impedance ratio difference at 5 kHz, the reactance ratio difference at 50 kHz, and the phase angle ratio difference at 50 kHz were 8.22 (screened patients = 48, screening rate = 16.16%), 13.78 (screened patients = 1, screening rate = 0.34%), 8.13 (screened patients = 52, screening rate = 17.50%), 17.59 (screened patients = 62, screening rate = 20.87%), and 19.73 (screened patients = 44, screening rate = 14.30%), respectively (Table 5).

## Verification data sets of subclinical standards

Table 6 provides a univariate analysis of the significant differences among the three groups. The ratio difference of ECW in the lymphedema breast cancer survivor group was higher than that in the other groups (healthy control group = 3.12±3.19, subclinical group = 18.25 ±9.23, and patient group = 44.05±35.72). The ECW/TBW of the unilateral arm in the

**Table 5. Inter-limbs differential analysis and screening rate.**

| Variables | Control group (N = 127) | The critical screening values (M ±2SD) | The number of screened patients | screening rate (%) |
|---|---|---|---|---|
| The ratio difference of ECW | 3.14±2.59 | 8.22 | 48 | 16.16 |
| The ratio difference of FFM | 3.47±5.26 | 13.78 | 1 | 0.34 |
| The ratio difference of Impedance in 5kHz | 3.07±2.58 | 8.13 | 52 | 17.50 |
| The ratio difference of Reactance in 50kHz | 5.89±5.97 | 17.59 | 62 | 20.87 |
| The ratio difference of Phase Angle in 50kHz | 7.37±6.31 | 19.73 | 44 | 14.81 |

*"Screened patients" were defined as those who fulfilled the criteria for subclinical lymphedema.

*"Screening rate" were defined as the number of screened patients /total number of patients*100%.

The critical screening values of various parameters were determined using the normal reference range and clinical judgment.

* ECW: Extracellular Water FFM: Fat Free Mass

lymphedema breast cancer group was also higher than that in the other groups (healthy control group = 0.379±0.037, subclinical group = 0.388±0.071, and patient group = 0.397 ±0.120). Analysis of variance revealed that participants in the healthy control group had a lower impedance ratio difference value at 5 kHz (health control group = 2.95±3.31, subclinical group = 37.70±26.31, and patient group = 43.55±33.87), the reactance ratio difference at 50 kHz (health control group = 7.34±8.65, subclinical group = 18.58±9.43, and patient group = 82.03±56.00), and the phase angle ratio difference at 50 kHz (health control group = 8.29±7.88), subclinical group = 22.66±15.12, and patient group = 28.87 ±16.13).

It was found that the subclinical standard we established can distinguish more than 90% (range, 90–100%) of the healthy participants from subclinical breast cancer survivors. Meanwhile, 60–90% of lymphedema breast cancer survivors can be detected by this standard.

**Table 6. Verification data sets of subclinic standard.**

| Index | reference range (M ±1.96SD) | screening rate (%) | healthy controls Mean(SD) | Subclinic group Mean (SD) | lymph-edema survivors Mean (SD) | F | P | The number (ratio) of healthy people excluded by this standard | The number (ratio) of patients meet this standard |
|---|---|---|---|---|---|---|---|---|---|
| The ratio difference of ECW | 8.22 | 16.16 | 3.12(3.19) | 18.25(9.23) | 44.05(35.72) | 28.112 | <0.001 | 28(93%) | 26(87%) |
| ECW/TBW of unilateral Arm | Right Arm = 0.3856 Left Arm = 0.3867 | 18.27 | 0.379(0.037) | 0.388(0.071) | 0.397(0.120) | 23.426 | <0.001 | 30(100%) | 25(83%) |
| The ratio difference of Impedance in 5 kHz | 8.13 | 17.50 | 2.95(3.31) | 37.70(26.31) | 43.55(33.87) | 44.222 | <0.001 | 28(93%) | 26(87%) |
| The ratio difference of Reactance in 50 kHz | 17.59 | 20.87 | 7.34(8.65) | 18.58(9.43) | 82.03(56.00) | 18.026 | <0.001 | 27(90%) | 27(90%) |
| The ratio difference of Phase Angle in 50 kHz | 19.73 | 14.81 | 8.29(7.88) | 22.66(15.12) | 28.87(16.13) | 35.973 | <0.001 | 28(93%) | 18(60%) |

## Discussion

Early recognition and intervention of lymphedema are very important for breast cancer survivors' health and quality of life. As a noninvasive measure, BI is well-suited for the early detection of lymphedema. In China, doctors recommend chemotherapy initiation ≤30 days after surgery. Patients underwent the measurement on the day before the first chemotherapy session, rather than after 3 months, compared with prior studies in Massachusetts [18]. This discrepancy was based on the measurement methods. The present study suggests that changes in BI can be detected prior to the onset of lymphedema symptoms. The detection of secondary lymphedema by bioimpedance will allow the identification of subclinical criteria based on predefined reference ranges. The established indicators of the subclinical criteria of lymphedema included the ECW ratio difference of > 16.16, the impedance ratio difference at 5 kHz of > 17.50, the phase angle ratio difference at 50 kHz of > 14.81, ECW/TBW of the right arm of > 0.3856, or ECW/TBW of the left arm of > 0.3867. Individuals who met any of the five criteria (Table 6) were determined to have subclinical impairment. These results demonstrate a relationship between subclinical lymphedema, invasive breast cancer, and lymph node removal. Breast cancer survivors who received intravenous chemotherapy before measurement had significantly changed assessment results. Therefore, it is important to exclude the effects of chemotherapy. Dissection/disruption of the axillary lymph nodes, radiotherapy, and chemotherapy are risk factors for the development of BCRL. In our study, we did not evaluate all the risks of BCRL but only the dissection/disruption of axillary lymph nodes.

A higher BMI is an important risk factor for lymphedema [19]. Our study revealed that breast cancer survivors had a significantly higher BMI, increased body fat mass, and a higher percentage of body fat. The frequency of BIA should vary with BMI, with increasing frequency at higher BMI. Regular evaluation and management of weight and body fat may reduce the risk of lymphedema. However, only 2.69–6.06% of patients had higher values than the cut-offs, and we were unable to screen patients using FFM or FFM% of the unilateral arm. Our study did not find a significant difference in interlimb FFM. Indicators of fat perform poorly in screening patients.

For the cellular water profile, ECW and ECW/TBW values associated with lymphedema are the focus of primary indexes to screen for patients with breast cancer [14, 19]. Abnormal accumulation of ECW is a fundamental cause of lymphedema [20]. Our data revealed that the value of ECW/TBW is substantially lower than that reported in other studies [21–24]. The different results of this study may be due to the different study populations. Regarding the different periods and features of the clinical and subclinical lymphedema groups, different criteria should be formulated for early diagnosis. It is worth noting that the two indicators of ECW entered the final subclinical diagnosis standard. Given the very early changes in ECW, our data highlight the possibility of screening patients with lymphedema in the early period after breast cancer surgery. For further data analysis, the ratio of ECW/TBW may be below the normal range (ECW/TBW of the right arm = 0.3856 or ECW/TBW of the left arm = 0.3867) in lymphedema patients with lower BMI (not mentioned in the results). However, this subset of patients can be screened using the indications of the ECW ratio difference. Few studies have focused on lymphedema patients with lower BMI, smaller arm circumference, and a lower ECW/TBW ratio, which might hinder the early identification and intervention of lymphedema. For these patients, the ratio difference in ECW is a better predictive index for lymphedema.

Additionally, BIA is widely used in a variety of diseases, including heart failure, sarcopenia, unilateral vestibular hypofunction, and metabolic syndrome but is not widely used in secondary lymphedema [24–26]. This finding of our study that the impedance of the low-frequency

current is a reliable index to evaluate lymphedema is consistent with previous studies at cancer centers and communities in New York City. However, our study population differed from the New York study population (race, patient characteristics, and treatment period). Moreover, the different results may be due to different BI devices and calculation methods [25]. Hence, the lack of an agreed-upon reference standard for screening and diagnosis may be a vital cause of the limited availability [26]. Our study revealed that bioelectrical information, including impedance, reactance, and body angle, is affected by mixed factors such as patient age, state of nutrition, and dominant or affected arms. Simply put, applying bioelectrical information of the unilateral arm is challenging when making a clinical or subclinical judgment of secondary lymphedema. However, bioelectrical information of the bilateral arms showed greater sensitivity and accuracy in screening patients with secondary lymphedema. Our study also confirmed that all values of BI, reactance, and body angle were similar between the right and left arms at different frequencies, except for the tendency of the right arm to have lower values. This discrepancy led us to consider the influence of the dominant arm on the bioelectrical measurements.

Volumetric assessment methods(circumference measurement, the water displacement technique and perometry) are the main diagnosis tool for BCRL. Previous study use, arm volume increased more than 3% corrected for body weight is widely used as a definition for early onset lymphedema. In addition, the threshold value of lymphedema relative volume (LRV) ≥5% was used to assess as a diagnostic threshold of clinic lymphedema based upon circumference measurements by water displacement of volume calculation. In case of circumference measurements, patients with circumferences measured at 4cm intervals starting from the middle fingertip [17, 27, 28]. However, all the volumetric assessment methods have disadvantages. BI are most appropriate or best fitting to diagnosis of subclinical BCRL [29, 30]. Previous studies have also applied the L-Dex ratio (measurement of extracellular fluid in the affected limb compared with the control arm) to diagnose secondary lymphedema [25]. However, the L-Dex ratio did not consider the inherent difference between the upper arms. Both mainstream diagnostic standards for lymphedema are based on changes in the upper limbs. Based on this theory, we built a dataset for the interlimb differential analysis. It is worth mentioning that the inherent differences in both upper arms should be considered.

Inherent differences in both upper arms evaluated in our study were defined as the normal range of the controls (healthy women). Hence, to assess the difference between the two arms, we evaluated 297 patients with breast cancer and 127 healthy women by including the ECW ratio difference, the FFM ratio difference, the impedance ratio difference at 5 kHz, the reactance ratio difference at 50 kHz, and the phase angle ratio difference at 50 kHz. Considering that only 0.34% of subclinical patients were screened using the FFM ratio difference, this index was excluded from the subclinical standard. Therefore, a higher ECW ratio difference, impedance ratio difference at 5 kHz, reactance ratio difference at 50 kHz, and phase angle ratio difference at 50 kHz are better screening tools for early recognition. There are several practical advantages of interlimb difference analysis. First, compared with previous standards, bioelectrical measurements are more precise and objective, and reduce measurement errors. Not only the actual difference but also the inherent difference of the upper arms in the assessment is considered. Second, we took advantage of the interlimb difference analysis to avoid distinguishing between dominant and nondominant arms and affected and unaffected arms. This can simplify the assessment, and the learning curve of the measurement is relatively short. Moreover, regardless of the type of bioelectrical device model used, we can compare different data.

Another fraction of our study verified the subclinical standard validity. This subclinical standard validity was assessed within two levels of clinical discrimination groups: the

subclinical lymphedema group vs. healthy control and subclinical lymphedema groups vs. the lymphedema group. Our findings revealed that for more than 90% of healthy control participants, the cut-off values of the subclinical standard were not accessible in clinical practice. Thus, the subclinical standard can distinguish subclinical lymphedema groups from healthy controls, with high discriminatory power. In our study, each of the five criteria was an independent index for screening patients with subclinical lymphedema. Between 60% and 90% of patients with lymphedema had higher values above the subclinical standard. However, 30 patients with lymphedema from our center were at different clinical stages, which inevitably had an impact on the change in ECW to some extent. Given a realistic situation, the recognition accuracy assessed from the subclinical standard is more optimistic. Therefore, the subclinical standard we established is a highly sensitive tool for the early detection of lymphedema, as shown in our study. Further studies need to use objective subclinical standards for taking special precautions and interventions. Our report showed that the incidence of subclinical lymphedema was 14.21% to 20.87%. Compared with the previous meta-analysis, the subclinical incidence in our study was consistent with the final prevalence results [8]. This suggests that the subclinical standard we established could be a successful standard for recognizing patients with subclinical lymphedema and could also be an effective indicator to predict secondary lymphedema. Our results also revealed that 14–20% of breast cancer survivors developed subclinical lymphedema very early after breast cancer surgery. Early prevention and intervention could deliver a larger medical cost-saving effect.

## Limitations

Our study has some limitations. This study is only a case–control trial, and we assessed the bioelectrical changes only once; we did not evaluate all the risks of BCRL except for dissection/disruption of axillary lymph nodes [31]. A prospective study would be help assess the impact of different treatment-related factors, such as invasive cancer diagnosis, dissection/disruption of axillary lymph nodes, chemotherapy, and radiation therapy. It can also evaluate the incidence of patients with subclinical lymphedema who will develop lymphedema [17]. Moreover, the cases only came from Western China, and a single-center study limits the generalizability of our study. Future research using a prospective multicenter large-sample cohort study design could improve the reliability of the findings. Furthermore, there were some between-group demographic (age and BMI) differences. We attempted to use group correction between the two groups by establishing a linear regression model. The statistical results showed that there was no need for correction between the groups using age and BMI.

## Conclusion

In clinical practice, early recognition and intervention for lymphedema in breast cancer survivors are necessary. Our study established a subclinical lymphedema standard for screening patients with lymphedema after surgery. There are five screening criteria, including the ECW ratio difference of > 16.16, the impedance ratio difference at 5 kHz of > 17.50, the phase angle ratio difference at 50 kHz of > 14.81, ECW/TBW of the right arm of > 0.3856, and ECW/TBW of the left arm of > 0.3867. Patients with breast cancer patients were at the subclinical lymphedema stage (International Society of Lymphology, stage 0) if their BI data satisfied any of the five criteria. The more criteria that are met, the greater the risk of developing lymphedema symptoms in the future. Our study implies that patients with breast cancer experience changes in BI in the early stages after surgery. These changes correlated with lymph node removal. Based on our results, early prevention and intervention are worthy of deep exploration.

## Author Contributions

**Conceptualization:** Linli Zhuang, Huaying Chen, Xuemei Zheng, Shaoyong Wu.

**Data curation:** Linli Zhuang, Huaying Chen, Shaoyong Wu.

**Formal analysis:** Shaoyong Wu.

**Investigation:** Youhui Yu, Lu Lan, Liang Xu, Jumei Xu, Hongying Fan.

**Methodology:** Shaoyong Wu, Youhui Yu, Lu Lan, Liang Xu, Jumei Xu, Hongying Fan.

**Project administration:** Huaying Chen, Youhui Yu, Lu Lan, Liang Xu, Jumei Xu, Hongying Fan.

**Resources:** Lu Lan, Liang Xu, Jumei Xu, Hongying Fan.

**Supervision:** Shaoyong Wu.

**Validation:** Linli Zhuang, Huaying Chen, Xuemei Zheng, Shaoyong Wu.

**Visualization:** Shaoyong Wu.

**Writing – original draft:** Linli Zhuang, Shaoyong Wu.

**Writing – review & editing:** Huaying Chen, Xuemei Zheng, Shaoyong Wu.

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
