## [Decision Letter · Decision Letter 0]

9 Jun 2022

PONE-D-22-02659Bioelectrical impedance analysis for early screening upper- limb  subclinic lymphedema: A case–control  studyPLOS ONE

Dear Dr. Zhuang,

Thank you for submitting your manuscript to PLOS ONE. After careful consideration, we feel that it has merit but does not fully meet PLOS ONE’s publication criteria as it currently stands. Therefore, we invite you to submit a revised version of the manuscript that addresses the points raised during the review process.

The reviewers both thought the manuscript addressed an important topic and could be important in evaluating patients for lymphedema.  However, they both had a number of comments which suggested further revisions to improve the value of the submission. Both reviewers indicated that more information regarding specific requirements for recruitment to the study should be clarified and explained since some of the requirements are at odds with previous published studies. The choice of the control population could also be better supported.  One of the reviewers has indicated that a fuller discussion of previous research would be beneficial in establishing the value of the current study.  The manuscript could also be improved by editing for English grammar and editing of the reference list for duplications. 

We look forward to receiving your revised manuscript.

Kind regards,

Robert M Lafrenie, PhD

Academic Editor

PLOS ONE

Journal Requirements:

Reviewers' comments:

Reviewer's Responses to Questions

**Comments to the Author**

1. Is the manuscript technically sound, and do the data support the conclusions?

Reviewer #1: Partly

Reviewer #2: No

2. Has the statistical analysis been performed appropriately and rigorously? 

Reviewer #1: I Don't Know

Reviewer #2: Yes

3. Have the authors made all data underlying the findings in their manuscript fully available?

Reviewer #1: Yes

Reviewer #2: Yes

4. Is the manuscript presented in an intelligible fashion and written in standard English?

Reviewer #1: No

Reviewer #2: No

5. Review Comments to the Author

Reviewer #1: In the field of lymphedema, a diagnostic tool that would be able to detect subclinical lymphedema would be very welcome. As the authors state Bio Impedance (BI) would be a potentional device/approach. However, current scientific literature is undecided on BI with different cut-off values amont different devices and populations. The current study tries to deal with some of the issues regarding BI. After thorough reading of the manuscript I have the following comments/concerns:

- the authors should consider English language editing

- it is nog clear why menstruating is an exclusion crterion, could a BI assessment not be postponed for several days

- In comparison with BI a reference with a 2cm- difference based on circumefereces was used to define lymphedema. As demonstrated by Armer, in 2005 already, the 2 cm diff definition is a wrong definition to be used in the assessment of lymphedema.

- For BI an approach of fasting and empty bladder is used => therefore this becomes a protocol that can only be used in the morning, limiting clinical application when patients are routinely seen on an ambulatory appointment.

- the author state that a skilled therapist examined body composition, what was assessed and how?

- A major concern is that as a reference a healthy group was used. This healthy sample is significantly different from the patient sample. At least I expected a matched sample to draw any conclusions.

- It is unclear how an interlimb difference can be caculated from a whol body BI. Additionally, a difference between dominant and non-dominant arms exist, this was found by the authors as well. How to deal with this in BI. The Methodology should elaborate on this, making section 3.5 better readable.

-Only 1 assessment was made to define subclinical lymphedema, this should be multiple times in follow up, as mentioned by the authors. Why not add prospective follow-up data to confirm the hypothesis that BI can be a diagnostic tool.

-Table 1 clearly describes the sign differences between groups

-Tables 2-5 cannot be interpret based upon the table alone, additonal information should be provided in a legend.

Reviewer #2: Although this is a very important topic, you seem to claim that this technique is novel, whereas there has been an extensive literature on the use of bioimpedance spectroscopy in the early detection of subclinical lymphoedema after breast cancer treatment (See examples of existing literature below). This has already addressed a number of issues which you raise as needing further exploration eg the effect of the dominant arm on bioimpedance values; the value of preoperative assessments followed by post operative measurements in demonstrating those who experience a change which then leads on to clinical lymphoedema; normative values and the use of a 2SD or 3SD cut offs (Fu 2013).

I think it would improve the paper significantly if you could review the existing literature and clarify what your findings add to current knowledge.

Specific points:

page 8 “Subjects”: second paragraph: Why did you exclude patients who had radiotherapy or chemotherapy? Radiotherapy and taxane chemotherapy are known to be risk factors for the development of breast cancer related lymphoedema.

Page 9: it is not clear here when the data were collected. You quote “about” the time of surgery but this seems to be post operatively and in the window before chemotherapy or radiotherapy treatment is given?

Page 9 evaluation criteria: In existing studies of the early detection of breast cancer related lymphoedema by limb volume measurements, a relative arm volume increase (compared with the contralateral arm) is commonly used. The two centimetres circumferential difference is on outmoded definition.

Page 9: the use of the term “screened patients” is not clear. Does it mean patients who fulfil the criteria of subclinical lymphoedema defined in your study?

Page 11 and 12: most of the existing literature uses the measurement of extracellular fluid (ECF) in the affected limb compared with the control arm, most commonly expressed as a ratio (L-Dex, in one manufacturer's device). This seems to that be validated by your study but you have also found some other measures which may be helpful. It will be important to compare these results with the existing literature and demonstrate if your other measures add anything to the existing knowledge.

page 14: I agree that a prospective multi centre study with a larger cohort would be helpful but you make no reference to the fact that these have been carried out using other bioimpedance devices for breast cancer related lymphoedema.

In your references, reference 13 duplicates reference 10; reference 22 duplicates reference 16; reference 21 duplicates reference 20 and reference 23 duplicates reference 15.

In the tables it would be important to define what is meant by “screened patients” as raised above under comments on page 9.

Example eferences to existing literature:

Rockson S, Keeley V, Kilbreath S, Szuba, A, Towers A. Cancer-associated secondary lymphedema. Nat Rev Dis Prim 2019; 5:22.

Fu M, Cleland C, Guth A, Kayal M, Haber J, Cartwright F, e al. L-Dex ratio in detecting breast cancer-related lymphedema: Reliability, sensitivity, and specificity. Lymphology 2013; 46:85–96.

Dylke, E. S. et al. Diagnosis of upper limb lymphedema: development of an evidence-based approach. Acta Oncol 55, 1477-1483, doi:10.1080/0284186X.2016.1191668 (2016).

Ward, L. C., Dylke, E., Czerniec, S., Isenring, E. & Kilbreath, S. L. Confirmation of the reference impedance ratios used for assessment of breast cancer-related lymphedema by bioelectrical impedance spectroscopy. Lymphat Res Biol 9, 47-51 (2011)

Bundred N, Foden P, Todd C, Morris J, Watterson D, Purushotham, A, et al. Increases in arm volume predict lymphedema and quality of life deficits after axillary surgery: A prospective cohort study. Br J Cancer 2020; 123:17–25.

6. PLOS authors have the option to publish the peer review history of their article (what does this mean?). If published, this will include your full peer review and any attached files.

Reviewer #1: No

Reviewer #2: No

---

## [Author Response · Author response to Decision Letter 0]

20 Jul 2022

Dear Dr. Lafrenie

Subject: Submission of revised paper PONE-D-22-02659

 Thank you for your email dated 9 June 2022 enclosing the reviewers’ comments. We have carefully reviewed the comments and have revised the manuscript accordingly. Our responses are given in a point-by-point manner below. Changes to the manuscript are shown in a separate file labeled “Revised manuscript with Track Changes”.

 We hope the revised version is now suitable for publication and look forward to hearing from you in due course.

Sincerely,

Shaoyong Wu

International lymphedema therapist

West China Hospital

Response to Reviewer 1: Thank you for your review of our paper. We have answered each of your points below.

1.the authors should consider English language editing

Response: Thanks very much for the suggestions from reviewer. Before resubmission, this manuscript edited and checked by a professional edition company (www.editage.com), which deals with English editing for non-native English researchers. We hope that our manuscript meets your requirements for publication now. We have also uploaded the proof of polishing the manuscript. 

2.it is not clear why menstruating is an exclusion criterion, could a BI assessment not be postponed for several days.

Response: Lymphadenectomy is the primary cause of breast and upper extremity lymphedema in patients with breast cancer. Our study is an exploratory study. Exploratory objective are to evaluate potential influence of lymphadenectomy for breast cancer survivors. For females, testing during menstruation influence BI outcome. Due to this constraint, we choose to menstruating is an exclusion criterion.

 In China, within 3 days post-consultation about treatment options, patients will receive first chemotherapy treatment. Patients spent these days in hospital. In fact, we chose the day before the data of first chemotherapy treatment to overcome the influence of chemotherapy. If BI assessment is postponed for several days, it's very likely that patients have received their first circle of chemotherapy. Chemotherapy is a major confounding factors in our study. Therefore, BI assessment could not be postponed for several days. In the end of the manuscript, we revised this paragraph(Page 5, Paragraph 2).

3.In comparison with BI a reference with a 2cm- difference based on circumefereces was used to define lymphedema. As demonstrated by Armer, in 2005 already, the 2 cm diff definition is a wrong definition to be used in the assessment of lymphedema.

Response: Thank you for your suggestion. In the article, our essay mentions the standard of 2cm- difference based on circumference. This standard is outdated. Your suggestion pointed out exactly where we had gone wrong. I have revised the contents of this part.(Page 7, Paragraph 3) Thank you very much. 

4.For BI an approach of fasting and empty bladder is used => therefore this becomes a protocol that can only be used in the morning, limiting clinical application when patients are routinely seen on an ambulatory appointment.

Response: Thank you, good question. Our study is an exploratory study. Hospitalized patients were recruited in order to facilitate longitudinal follow-up. It is also easy to implement and measure in practice. In fact, no one received initial chemotherapy as an outpatient in China. Moreover, The result of BI may be influenced by a multiplicity of different factors. To avoid possible influences of eating and drinking, an approach of fasting and empty bladder is used in our study. Further research regarding underlying influence of times(morning or afternoon) and meals is warranted. It would be a good idea to assess the feasibility of clinical application of outpatient. Thank you very much.

5.the author state that a skilled therapist examined body composition, what was assessed and how?

Response: Thank you for your question. I am very sorry that this part was not clear in the original manuscripts, I should have explain that our assessment will cost 5-10 minutes before measurement. A skilled therapist in our study assessed several aspects before measurement, such as whether the inclusion and exclusion criteria were met, whether skin changes occurred, whether limb pain or discomfort existed, whether the patient experienced upper extremity swelling or nonpitting edema, and whether the range of motion was restricted. BIA data were collected by a trained therapist. Implementation of guideline-recommended assessment in our study is appropriate for breast cancer patients. I have revised the content of this part. (Page 6, Paragraph 1)

6.A major concern is that as a reference a healthy group was used. This healthy sample is significantly different from the patient sample. At least I expected a matched sample to draw any conclusions.

Response: Thank you for your question. Our group member give serious consideration to this matter. We'd like to know the differences between groups in one of primary outcome measure（the ratio differences of impedance in 5KHz). However, there were some between-group demographic(age and BMI) differences. we establish the linear regression model by correcting demographic(age and BMI) differences. In fact, age and BMI were excluded variables. Only groups is stepwised entrying into the linear regression equations. These statistics show that there's no need for correction between groups using age and BMI. Therefore, we do not use the method of propensity score matching. For details, see below:

7.It is unclear how an inter-limb difference can be calculated from a whole body BI. Additionally, a difference between dominant and non-dominant arms exist, this was found by the authors as well. How to deal with this in BI. The Methodology should elaborate on this, making section 3.5 better readable.

Response: Thank you for your question. I am very sorry that this part was not clear in the original manuscripts, I should have explain that data collected form bioimpedance device includes five parts: left arm part, the right arm part, trunk part, left leg part and right leg part. All parts of data are numerical-type variables. Therefore, an inter-limbs difference can be calculated between the values of left and right arm in the same index. 

 In fact, the difference between both arms exists not only in the dominant and nondominant arms but also in the proximal and distal arms (according to the distance between the breast cancer location and arm). Our study showed that differences between arms exist in breast cancer survivors, which has not been suggested by previous research. As an exploratory study, the difference between both arms in the healthy group will be used to provide a benchmark that uses two standard deviation cut-offs for screening breast cancer survivors. Briefly, dominant arm is indistinguishable from the non-dominant arms. The concept(the difference between both arm) is better for us to assess the change of arm. Our study contrasted the difference between both arm in healthy group and in breast cancer participants.

 Moreover, I have revised the content of Methodology, especially the content of section 3.5. (Page 11, Paragraph 2)

8.Only 1 assessment was made to define subclinical lymphedema, this should be multiple times in follow up, as mentioned by the authors. Why not add prospective follow-up data to confirm the hypothesis that BI can be a diagnostic tool.

Response: On the one hand, follow-up assessment may be affected because of follow-up treatment(chemotherapy and radiotherapy). The results of follow-up assessment does not represent this assessment. It is not necessary connection between follow-up and this measurement.On the other hand, this study formed part of a larger programme of research. Our research is a prospective cohort study. We will implement multiple follow-ups. It'll take 2 year, at the very least. That is the reason why we not add prospective follow-up data to confirm the hypothesis that BI can be a diagnostic tool.Therefore, your suggestion is the question which we will solve in the future.

 However, BI is the diagnostic aids recommended by Clinical Practice Guideline From the Academy of Oncologic Physical Therapy of APTA. More interestingly, we find the possible change at the first follow-up after the surgery of breast cancer. This change closely related to breast cancer surgery which is one of the most common causes of upper extremity lymphedema. We are recognized that the possible change may impact the development of lymphedema. 

 In the end of the manuscript, we revised this paragraph and we explain the reason in the Materials and methods part. (Page 2, Paragraph 2)

9.-Table 1 clearly describes the sign differences between groups

10.-Tables 2-5 cannot be interpret based upon the table alone, additional information should be provided in a legend.

Response: Thank you for your suggestion. we had provided additional information in the article and in a legend. Thank you. I learn a great deal from your propose. I have revised the content of this part.(Table1-5)

Special thanks to you for your good comments.

Response to Reviewer2: Thank you for your review of our paper. We have answered each of your points below.

1.Although this is a very important topic, you seem to claim that this technique is novel, whereas there has been an extensive literature on the use of bioimpedance spectroscopy in the early detection of subclinical lymphoedema after breast cancer treatment (See examples of existing literature below). This has already addressed a number of issues which you raise as needing further exploration eg the effect of the dominant arm on bioimpedance values; the value of preoperative assessments followed by post operative measurements in demonstrating those who experience a change which then leads on to clinical lymphoedema; normative values and the use of a 2SD or 3SD cut offs (Fu 2013).

Response: Thank you for your question. We read the literature you recommend carefully. We believe your advice will improve the paper significantly. I have reviewed the existing literature and then revised the contents of this article. In fact, this technique is not novel. Yet, the correlation between invasive cancer diagnosis, dissection/disruption of axillary lymph nodes and secondary lymphedema are not yet understood and assessed. It is core to our study. Moreover, there are many types of bio-impedance devices that are routinely used in clinical practice around the world. The absolute values therefore may not be directly comparable. So it's an exploratory experiment to make an attempt on setting up a new way to make data from different study comparable. (Page 2, Paragraph 2)

2.page 8 “Subjects”: second paragraph: Why did you exclude patients who had radiotherapy or chemotherapy? Radiotherapy and taxane chemotherapy are known to be risk factors for the development of breast cancer related lymphoedema.

Response: Thank you for your question. Dissection/disruption of axillary lymph nodes, radiotherapy and chemotherapy are the risk factors for the development of breast cancer related lymphedema. In our study, we did not evaluate all the risk of breast cancer related lymphedema, but only dissection/disruption of axillary lymph nodes. Therefore, we exclude patients who had radiotherapy or chemotherapy. These treatment will cause major effects on results. (Page 5, Paragraph 2)

3.Page 9: it is not clear here when the data were collected. You quote “about” the time of surgery but this seems to be post operatively and in the window before chemotherapy or radiotherapy treatment is given?

Response: Thank you for your question. I am very sorry that this part was not clear in the original manuscripts, I should have explain that surgical times were recorded as one of surgical data. And we chose the day before the data of first chemotherapy treatment to overcome the influence of chemotherapy. I have revised the content of this part.(Page 5, Paragraph 2)

4.Page 9 evaluation criteria: In existing studies of the early detection of breast cancer related lymphoedema by limb volume measurements, a relative arm volume increase (compared with the contralateral arm) is commonly used. The two centimetres circumferential difference is on outmoded definition.

Response: Thank you for your suggestion. In the article, our essay mentions the standard of 2cm- difference based on circumference. This standard is outdated. Your suggestion pointed out exactly where we had gone wrong. I have revised the contents of this part(Page 7, Paragraph 1). Thank you very much. 

5.Page 9: the use of the term “screened patients” is not clear. Does it mean patients who fulfil the criteria of subclinical lymphoedema defined in your study?

Response: Thank you for your question. I am very sorry that this part was not clear in the original manuscripts. The term “screened patients” is defined as patients who fulfil the criteria of subclinical lymphedema in our study. I will give a clear definition of this word in our article.(Page 7, Paragraph 1)

6.Page 11 and 12: most of the existing literature uses the measurement of extracellular fluid (ECF) in the affected limb compared with the control arm, most commonly expressed as a ratio (L-Dex, in one manufacturer's device). This seems to that be validated by your study but you have also found some other measures which may be helpful. It will be important to compare these results with the existing literature and demonstrate if your other measures add anything to the existing knowledge.

Response: Thank you for your suggestion. I believe that your suggestion is important and I'll follow any advice you give me. We added the part that compare our results with the existing literature.(Page 15, Paragraph 2) Thank you very much. 

7.page 14: I agree that a prospective multi centre study with a larger cohort would be helpful but you make no reference to the fact that these have been carried out using other bioimpedance devices for breast cancer related lymphedema.

Response: Thank you for your question. The main risk factors for breast cancer-associated lymphedema (BCAL) include invasive cancer diagnosis, dissection/disruption of axillary lymph nodes, radiation therapy, local infection, and obesity, but other factors may also contribute. A prospective multi-centre study with a larger cohort would be helpful to assess the impact of different treatment related factors. We added the part that compare our results with the existing literature to explain the difference. (Page 12-17)

8.In your references, reference 13 duplicates reference 10; reference 22 duplicates reference 16; reference 21 duplicates reference 20 and reference 23 duplicates reference 15.

Response: Thank you for your question. We are very sorry for our negligence on references. I have revised the contents of this part. (Page 20-23)

9.In the tables it would be important to define what is meant by “screened patients” as raised above under comments on page 9.

Response: Thank you for your question. The term “screened patients” is defined as patients who fulfil the criteria of subclinical lymphedema in our study. I will give a clear definition of this word in the table.(Page 7, Paragraph 1)

 We tried our best to improve the manuscript and made some changes in the manuscript. These changes will not influence the content and framework of the paper. And here we did not list the changes but marked in red in revised paper.

We appreciate for Editors/Reviewers’ warm work earnestly, and hope that the correction will meet with approval.

 Once again, thank you very much for your comments and suggestions.

---

## [Decision Letter · Decision Letter 1]

27 Jul 2022

PONE-D-22-02659R1Bioelectrical impedance analysis for early screening of upper limb subclinical lymphedema: A case–control studyPLOS ONE

Dear Dr. Wu,

Thank you for submitting your manuscript to PLOS ONE. After careful consideration, we feel that it has merit but does not fully meet PLOS ONE’s publication criteria as it currently stands. Therefore, we invite you to submit a revised version of the manuscript that addresses the points raised during the review process.

In going through the revised version of the manuscript, the reviewer has indicated that the manuscript has been substantially improved but that a few points arise that need to be addressed prior to publication.  For example, the definition of lymphedema is based on an older definition, some of the references may not be the most appropriate, and a picture of the apparatus would be beneficial. 

We look forward to receiving your revised manuscript.

Kind regards,

Robert M Lafrenie, PhD

Academic Editor

PLOS ONE

Journal Requirements:

Reviewers' comments:

Reviewer's Responses to Questions

**Comments to the Author**

1. If the authors have adequately addressed your comments raised in a previous round of review and you feel that this manuscript is now acceptable for publication, you may indicate that here to bypass the “Comments to the Author” section, enter your conflict of interest statement in the “Confidential to Editor” section, and submit your "Accept" recommendation.

Reviewer #1: All comments have been addressed

2. Is the manuscript technically sound, and do the data support the conclusions?

Reviewer #1: Yes

3. Has the statistical analysis been performed appropriately and rigorously? 

Reviewer #1: Yes

4. Have the authors made all data underlying the findings in their manuscript fully available?

Reviewer #1: No

5. Is the manuscript presented in an intelligible fashion and written in standard English?

Reviewer #1: Yes

6. Review Comments to the Author

Reviewer #1: The authors have done a great job in revising their manuscript with regards to the English language and the topics raised by the reviewers.

Some comments still remain:

Line 135-136 The authors now state that the 200ml or 10% vol diff are today's standard definition for BCRL. These definitions are also outdated. Today we use, 3% vol change corrected for body weight as a definition for early onset lymphedema. In patients suffering from edema we use <5% between 5-10% and >10% vol diff assessed either by water discplacment of volume calculation based upon circumference measuerements. In case of circumference measurements, measeurments are taken every 4 cm (4CM increments). The refs 30-33 that are used to motivate the 10% or 200ml are not approprioate as they are refs concerning BIA. Add the original research (and as mentioned in previous revies, ARMER et al have investigated these definitions)

Additional refs that are lacking are: PMID: 33646139 and PMID: 33245458

Line 96-97 is repetition of the eligibility criteria lines 102-105

I still lack a piture of the BIA setup (preferably demonstrated on a patient)

7. PLOS authors have the option to publish the peer review history of their article (what does this mean?). If published, this will include your full peer review and any attached files.

Reviewer #1: No

---

## [Author Response · Author response to Decision Letter 1]

16 Aug 2022

Response to Reviewer 1: Thank you for your review of our paper. We have answered each of your points below.

1.Line 135-136 The authors now state that the 200ml or 10% vol diff are today's standard definition for BCRL. These definitions are also outdated. Today we use, 3% vol change corrected for body weight as a definition for early onset lymphedema. In patients suffering from edema we use <5% between 5-10% and >10% vol diff assessed either by water discplacment of volume calculation based upon circumference measuerements. In case of circumference measurements, measeurments are taken every 4 cm (4CM increments). The refs 30-33 that are used to motivate the 10% or 200ml are not approprioate as they are refs concerning BIA. Add the original research (and as mentioned in previous revies, ARMER et al have investigated these definitions). Additional refs that are lacking are: PMID: 33646139 and PMID: 33245458

Response: Thank you for your suggestion. In the article, our essay mentions the standard of 200ml or 10% volume difference. This standard is outdated. Your suggestion pointed out exactly where we had gone wrong. I have revised the contents of this part.(Line 135-137, 318-327) Thank you very much. 

I've read all the literature recommended by reviewer and then revised the contents of this article. (Line 476-482) We are very sorry for our negligence on references. Thank you very much. 

2.Line 96-97 is repetition of the eligibility criteria lines 102-105

Response: Thanks very much for the suggestions from reviewer. I have revised the contents of this part.(Line 96-97, 104) Thank you very much. 

3.I still lack a piture of the BIA setup (preferably demonstrated on a patient)

Response: Thank you for your question. As shown in the picture 1, we use InBody 770 multifrequency BI device (InBody 770, Cerritos, CA, USA) to measure and assess BIA of breast cancer survivors. Picture 2 present this BIA setup demonstrated on a patient.

Picture 1 InBody 770, Cerritos, CA, USA

Picture 2 BIA setup (demonstrated on a patient)

 We tried our best to improve the manuscript and made some changes in the manuscript. These changes will not influence the content and framework of the paper. And here we did not list the changes but marked in red in revised paper.

We appreciate for Editors/Reviewers’ warm work earnestly, and hope that the correction will meet with approval.

Once again, thank you very much for your comments and suggestions.

---

## [Decision Letter · Decision Letter 2]

31 Aug 2022

Bioelectrical impedance analysis for early screening of upper limb subclinical lymphedema: A case–control study

PONE-D-22-02659R2

Dear Dr. Wu,

We’re pleased to inform you that your manuscript has been judged scientifically suitable for publication and will be formally accepted for publication once it meets all outstanding technical requirements.

Kind regards,

Robert M Lafrenie, PhD

Academic Editor

PLOS ONE

Additional Editor Comments (optional):

Reviewers' comments:

Reviewer's Responses to Questions

**Comments to the Author**

1. If the authors have adequately addressed your comments raised in a previous round of review and you feel that this manuscript is now acceptable for publication, you may indicate that here to bypass the “Comments to the Author” section, enter your conflict of interest statement in the “Confidential to Editor” section, and submit your "Accept" recommendation.

Reviewer #1: All comments have been addressed

2. Is the manuscript technically sound, and do the data support the conclusions?

Reviewer #1: Yes

3. Has the statistical analysis been performed appropriately and rigorously? 

Reviewer #1: Yes

4. Have the authors made all data underlying the findings in their manuscript fully available?

Reviewer #1: Yes

5. Is the manuscript presented in an intelligible fashion and written in standard English?

Reviewer #1: Yes

6. Review Comments to the Author

Reviewer #1: The authors have revised the manuscript appopriately to the comments made.

The pictures of the BIA device were included in the response letter. Please make sure they are published with the manuscript as well.

7. PLOS authors have the option to publish the peer review history of their article (what does this mean?). If published, this will include your full peer review and any attached files.

Reviewer #1: No

---

## [Editor Report · Acceptance letter]

6 Sep 2022

PONE-D-22-02659R2 

Bioelectrical impedance analysis for early screening of upper limb subclinical lymphedema: A case–control study 

Dear Dr. Wu:

I'm pleased to inform you that your manuscript has been deemed suitable for publication in PLOS ONE. Congratulations! Your manuscript is now with our production department. 

Kind regards, 

on behalf of

Dr. Robert M Lafrenie 

Academic Editor

PLOS ONE